# University Teaching Planning in Times of COVID-19: Analysis of the Catalan Context and Proposal for a Future Model from ESIC Business and Marketing School Experience

Javier Bustos Díaz [†] [iD], Teodor Mellen Vinagre *,[†] [iD] and Ruben Nicolas-Sans [†] [iD]

ESIC Business and Marketing School, ESIC Campus Universitario, Passeig Santa Eulàlia, 2,
08017 Barcelona, Spain; javier.bustos@esic.edu (J.B.D.); ruben.nicolas@esic.edu (R.N.-S.)
* Correspondence: teodor.mellen@esic.edu; Tel.: +34-931-170-784
† These authors contributed equally to this work.

**Abstract:** On 14 March 2020, Spain came to a standstill and the movement of people was restricted with the publication of Royal Decree (RD) 463/2020 and the education sector had to reinvent itself with new rules and procedures. The purpose of this paper is to analyse the impact of the different government regulations and their effect on university teaching planning. Since the approval of RD 555/2020, educational competences were returned to the autonomous communities and each of them implemented different public policies. We will analyse the specific impact on teaching planning and models applied in Catalonia and their development, which vary according to the evolution of the number of COVID-19 infections and which has been marking the political, economic and social agenda since the beginning of the pandemic. The university has moved from teaching in a face-to-face environment to a virtual or semi-virtual or blended learning environment. This change implies a paradigmatic transformation in communication, education, organisation, evaluation and planning, among others. At ESIC Business & Marketing School (ESIC) these pedagogical paradigm shifts have contributed to transform the learning processes in a context of pedagogical innovation.

**Keywords:** teaching planning; university; COVID-19; Catalonia

## 1. Introduction

During the time of this pandemic, numerous studies have proliferated on the effects of the COVID-19 crisis on the educational process [1–5] that emphasise the different factors that give content to the academic perspective, such as teaching assessment methods [6], teaching activity [7] and digital transformation in the classroom [8].

The first consequence of the health crisis caused by Covid-19 was the obligation to cancel face-to-face classes at all educational levels and to adapt teaching to a non-face-to-face scenario as a matter of urgency. These circumstances made it necessary to develop different teaching strategies to those usually employed in a very short period of time, and these have posed an enormous challenge for teachers and students.

In this new scenario, the urgency of teaching in a blended or non-presential way has been triggered, making it necessary to readapt and redefine all these educational methodologies and tools under these new terms. As a result, a wide range of resources have emerged for the delivery of this new distance or online learning.

As Fanelli, Marquina and Rabossi point out, it is clear that, although we are witnessing a positive learning process in terms of adaptation, "they are unlikely to bring about a 180-degree change" [9]. In universities, virtuality is likely to complement face-to-face learning, but without replacing it.

It can be said that we are witnessing a change in the educational paradigm that can be summarised in five main issues:

1. From the face-to-face classroom to the virtual university classroom in the context of a pandemic.
2. From the lecture model, in which communication flows vertically (from top to bottom) to another model of multiple interaction with different directions in educational interaction.
3. New horizontal communications through multiple technological channels.
4. New interactions technologically mediated by ICT that require a modification of the content being taught, the form of message transmission and new assessment strategies.
5. New models of participation in the classroom.

In this new scenario, university teaching should continue to provide teaching adapted to the knowledge society, promoting active, reflective and critical learning based on collaboration between teachers and students [10] and also provide teaching adapted to the knowledge of society [11].

These changes in teaching methodology resulting from the pandemic situation are in addition to those produced by the implementation of the European Higher Education Area (initiated in 1999 with the Bologna Declaration),in which the main transformations in methodological processes allowed a shift from content-based learning to a different type of learning based on competences, which in the latter case implies the necessary greater involvement of students [12,13]. In the specific case of university education "the competences that students have to develop, apart from those established by the Spanish Qualifications Framework for Higher Education, are of two types: generic or transversal and specific." [14].

Authors such as Villa [15] argue that the application of this competency-based model is a system generally accepted by the university community. However, its actual implementation has not been fully carried out, as "it requires strong human innovation (teacher training, innovative attitudes, teacher coordination, teamwork and greater collegiality, clear and decisive leadership of academic leaders...), and technological innovation (digital platform resources, wifi, suitable classrooms, technological support . . . )" [15].

## 2. Theoretical Framework

Authors such as Assunção and Gago point out that the closure of teaching institutions that has given way to an online or hybrid model, depending on the occasion, can be seen "as opportunities to learn and reshape traditional roles and practices." [16].

In this sense, as De Vincenzi points out, "this health crisis has provided teaching with an enormous opportunity to rethink the way in which the teaching process is conceived and exercised, to reflect on where and how we teach and to improve both dimensions of educational practice when we return to face-to-face teaching." [17].

The unprecedented nature of this health crisis in our recent history has led to heterogeneous responses by education providers to the multiple challenges created by the pandemic in higher education settings [18].

According to UNESCO [19] in some 185 countries schools have been closed due to the COVID-19 pandemic and millions of students have experienced disruptions in their education. Naciri [20] explains that thousands of universities and colleges around the world had closed their institutions to encourage social distancing measures to prevent the spread of the virus.

The arrival of the pandemic produced a paradigm shift, both in Spain and in the rest of the world, radically changing the teaching landscape, making it necessary and mandatory to move to fully online teaching from one day to the next in order to be able to continue classes and finish the academic year [21] "The pandemic situation has begun to accelerate the use and appropriateness of online education, which has highlighted the familiar problem of unequal access to technology and its effective application." [22].

Many teachers had to adapt and learn how to handle various videoconferencing, video recording and editing programmes, as well as how to take full advantage of the teaching platforms [6]. However, "Were universities ready to move from face-to-face to virtual teaching? Yes and no." [22].

As Baladron, Correvero and Manchado [8] point out, the reports on the use of ICTs in Spanish universities were positive, however, as these same authors point out, the pandemic has highlighted the lack of technological resources by many university institutions and the lack of training of teaching staff in digital terms.

On the other hand, the abrupt context experienced in Spain must be assumed, which, as in the rest of the countries of the world, must place value on the "many examples of voluntarism on the part of teachers [who] converted spaces in their homes into improvised classrooms, while, forced by circumstances, they entered into the environment of educational technologies." [23].

Quintana [24] argues that "quality exists in a college or university to the extent that adequate and appropriate resources are successfully directed to the task of achieving outcomes related to the institution's mission, and to the extent that the college or university's programmes make a significant and positive related difference.".

## 3. Methodology

In today's university there is a clear need to create dynamic teaching projects that are not dependent on the student's presence in the classroom.

To this end, this text is based on a qualitative methodological approach, with the aim of understanding and interpreting complex phenomena rather than quantifying them [25] in order to understand an entity, vital phenomenon or specific situation as deeply as possible [26,27]. Therefore, the emphasis is placed on social phenomena developed in their own natural environment of production and in which the subjective aspects of human behaviour are analysed above the objective characteristics, exploring and manifesting the subjective meaning that the human actor attaches to his or her action [28]. This perspective encompasses data collection techniques such as participant observation, discussion groups, structured or semi-structured interviews, life histories and content analysis, among others. Qualitative methodology provides comprehensive information on social phenomena with greater richness and depth than that obtained through quantitative techniques [28].

Specifically, this article is based on the qualitative content analysis of the main regulations published for the university from the implementation of the state of alarm until the return to institutional normality. This is an analysis of the regulatory documents using this data collection technique through a systematic and objectified methodology because it uses "procedures, variables and categories that respond to study designs and analysis criteria, defined and explicit." [29].

To this end, the work is divided into two sections that shape the results: firstly, the analysis of the different national regulations applied as a consequence of the declaration of the state of alarm, and secondly, the analysis of those regulations of an autonomous nature after the return of competences to the autonomous communities. Finally, the article proposes a teaching project which takes into account the dynamics obtained from the first phase.

1. Phase 1. Collection of the regulations under study and establishment of categories.
2. Phase 2. Comparison of COVID-19 regulations and their red application black on the university. In this stage of the study, the following laws and measures are compared: RD 463/2020; Ministry of Universities Regulations of 8 April and 10 June; CRUE Regulations of 16 April.
3. Phase 3. Analysis of the Sectoral Plan for Universities in Catalonia. This stage of the study focuses on the evolution of the different models applied in Catalonia, as well as their description.

After examining the different elements that have affected the university community in Spain, and in Catalonia in particular, the model applied by ESIC Business and Marketing School (ESIC) is described as a case study.

This stage of the study shows how the different initiatives adopted by this education centre, which go beyond face-to-face or remote teaching, present an innovative approach capable of offering a novel response to the situation caused by the health pandemic.

Ultimately, a proposal is put forward for a teaching project capable of transforming and adapting to future situations marked by uncertainty.

These are initiatives that belong to the so-called active learning strategies through which university teaching is adapted to new contexts marked by the need for training that follows a new methodology. These are active learning strategies such as flipped classroom and collaborative and cooperative learning.

## 4. Results

### 4.1. Comparison of COVID-19 Regulations and Their Effect on Universities

University teaching has undergone various transformations as a result of the application of suggestions and regulations to deal with this exceptional situation, such as those issued by the Ministry of Universities [30] and the CRUE [31] to deal with teaching evaluations during the confinement period. Similarly, since 18 June 2020, the Generalitat de Catalunya, making use of its legislative powers, has drawn up a Sectoral Plan for Universities also aimed at responding to the situation resulting from the pandemic caused by COVID-19. A summary of these regulations is set out in the following Table 1.

**Table 1.** Overview of key legislative initiatives in higher education to address the health pandemic.

| Law/Regulation | Area | Date | Type of Teaching | Type of Evaluation | Allows for Classroom Practice |
|---|---|---|---|---|---|
| RD 463/2020 | National | 14 March | Online | Online | No |
| Ministry of Universities | National | 8 April | Adapted attendance | Online | No |
| CRUE | National | 16 April | Online | | Yes |
| Ministry of Universities | National | 10 June | Adapted attendance | | No |
| Sectoral Plan for Universities Catalonia | National | | Hybrid | Face-to-Face | Yes |

Teaching is going through a period of constant transformation and adaptation due to the situation generated by the SARS-CoV-2 virus. With regard to the application and effect of the different laws, regulations and work plans on the university sector, which have been drawn up by the different competent institutions, we can distinguish two main stages:

1. The first phase is determined by RD 463/2020 of 14 March, which declares a state of alarm for the management of the health crisis situation caused by COVID-19.
2. The second phase begins on 18 June, when Catalonia recovered its competences and starts to draw up the Sectoral Plan for the University of Catalonia.

4.1.1. Comparison of COVID-19 Regulations: First Stage and the Consequences of RD 463/2020

According to data provided by the Ministry of Health, the last day on which Spain had 0 SARS-CoV-2 infections was 24 February 2020. Eighteen days later, the number of daily cases diagnosed in Spain (Figure 1) was 1708 persons. In response to the high rate of new infections, the Spanish Prime Minister, Pedro Sánchez, announced Royal Decree 463/2020, which brought into force the state of alarm and put an end, for the first time in Spain's democratic history, to the free movement of people.

Article 9 of this Royal Decree establishes the measures of containment in the field of education and training in which the following precautionary guidelines are put into effect:

1. Face-to-face educational activity is suspended in all centres and stages, cycles, grades, courses and levels of education referred to in Article 3 of Organic Law 2/2006, of 3 May on Education, including university education, as well as any other educational or training activities provided in other public or private centres.
2. During the period of suspension, educational activities shall be maintained through distance and online modalities, whenever possible.

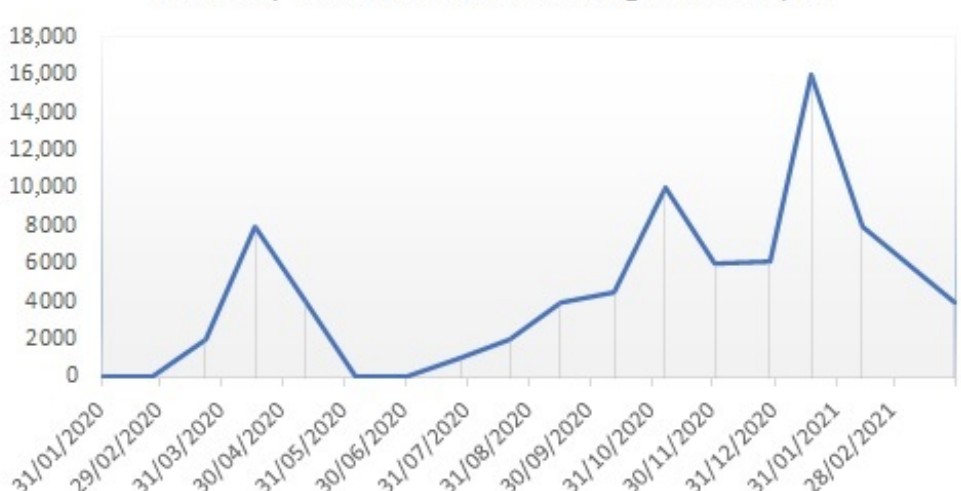

**Figure 1.** New cases of coronavirus daily diagnosed in Spain (31 January 2020–28 February 2021). Source: EPDATA [32].

Since the publication of the decree until the end of the 2019–2020 academic year, the online mode was maintained throughout Spain. At this point, three questions arise: was the Spanish university prepared to take on such an advanced technological leap, was there a real transformation from face-to-face teaching to the online model, or was the same traditional model transferred to a digital environment? And most importantly, were there any guarantees on how to assess students online?

In order to answer the latter question in particular, the Ministry of Universities produced and disseminated on 8 April 2020 the Report on initiatives and tools for online university assessment in the context of Covid-19. This document presents the following types of exams: questionnaires and short questions, oral exams and presentations, student work, projects and portfolios. Furthermore, in addition to these assessment procedures, suggestions for carrying out these practices were provided and are shown in Table 2.

In order to deepen and apply the procedures and premises to be introduced in the development of the different types of assessment systems, the CRUE published the Report on Non-Personal Assessment Procedures. Study of the Impact of their Implementation in Spanish Universities and Recommendations. This was a more complete manual in which three main blocks could be found:

1. Introduction and context: the situation experienced after the first month of the pandemic and the changes experienced in Spanish universities in global terms are analysed.
2. General aspects: general and regulatory aspects, the different methodologies envisaged and the applicable technology. General issues of security and portability of information and availability of IT systems throughout the process are also covered.
3. Design of non-face-to-face assessment procedures: the summary of this thematic block is shown in Table 3.

**Table 2.** Types of online assessments during the state of alarm.

| Typology | Technology | Recommendations for Evaluation | Systems to Ensure Authorship |
|---|---|---|---|
| Questionnaire and short questions | Moodle Classroom | Modify the typology of questions. | Online proctored exams |
| | Test Online | Conduct exams with randomised questions. | Supervision live online |
| | EvalBox TestWe | Select an appropriate test duration. | Without supervision |
| Oral exam and presentations | JITSI | Identify students by means of their student card. | Ensure that in the test all files on the student's computer are closed using the "screen sharing" system. |
| | Whereby | The tool must be validated in terms of IT security and must have an audit to ensure this. | Adjust the perspective of your camera to be able to see the student's face, the desk and the media he/she is handling during the conversation/test resolution. |
| | Video Etherpad | | If the examination is recorded, the videos must be stored for as long as necessary. |
| | E-Oral Microsoft Teams WebEx meetings (CISCO) Zoom Skype | | |
| Work, project and portfolio development | | Include among the objectives of the work reflections that make it difficult to copy from a repository. | Turnitin |
| | | Combine the submission of the paper with an oral presentation or interview to demonstrate mastery of the subject matter. | Urkund |

**Table 3.** Types of evaluations for a non-face-to-face context.

| Test | Purpose | Data | Treatment |
|---|---|---|---|
| Common | Controlling arbitrary or illicit actions | Common: Teacher is responsible for student identification | Depends on each exam |
| 1. Oral exam | Test procedure | Image and voice | Recording |
| 2. Open written exam | Anti-plagiarism | Student data | Data analysis |
| 3. Objective exam. | Anti-plagiarism | Student data | Data analysis |
| 4. One minute paper | Anti-plagiarism | Student data | Data analysis |
| 5. Academic work | Anti-plagiarism | Student data | Data analysis |
| 6. Concept maps | Anti-plagiarism | Student data | Data analysis |
| 7. Reflective journal | Anti-plagiarism | Student data | Data analysis |
| 8. Portfolio | Anti-plagiarism | Image and voice Details of participants | Recording |
| 9. Observation | Registration of the test | Image and voice | Recording |
| 10. Projects | Anti-plagiarism | Image and voice Details of participants | Recording |
| 11. Problems/Cases | Anti-plagiarism | Student data | Data analysis |

Table extracted from the Report on Non-presential Assessment Procedures. Study of the Impact of their Implementation in Spanish Universities and Recommendations.

From this first stage marked by the abrupt change from a face-to-face model to an online mode, it is necessary to try to provide universities with the means to carry out their evaluation task in a secure way.

4.1.2. Towards a Blended Model. Recommendations for the Academic Year 2020–2021 at National Level

Shortly afterwards, on 10 June 2020, in order to prepare for the planning of the 2020–2021 academic year, a report entitled Recommendations of the Ministry of Universities to the University Community to Adapt the 2020–2021 Academic Year to an Adapted Face-to-Face Course was developed.

This document contains various guidelines to be followed on the basis of the recommendations already formulated by the Ministry of Health, in which the following aspects should be highlighted:

1.  Systems to ensure authorship.

$$Occupancy = \frac{\text{Number of students enrolled in the activity}}{\text{Capacity of the installation taking into account the 1.5 metre separation between occupants}}$$

2.  In the same way, each university should adapt this scenario to its own reality on the basis of the following formula:

$$Occupancy = \frac{\text{Number of students enrolled in the activity}}{\text{Real capacity of the installation}}$$

In addition to recommendations on how to avoid doubling classes by groups in order to maintain face-to-face teaching, the document clarifies that "Each university, in close collaboration with its competent educational administration, will establish before the beginning of the academic year 2020–2021 a contingency plan that will allow, in case the health situation so requires, a massive and immediate change to an online teaching system."

Finally, hygiene measures are established under four headings: contact limitation, personal prevention measures, case management and cleaning and ventilation.

Subsequently, on 18 June 2020, the Generalitat de Catalunya recovered its autonomous powers and drew up the Sectoral Plan for Universities to respond to the situation resulting from the pandemic caused by COVID-19.

*4.2. Comparison of Regulations COVID-19: Sectoral Plan for Universities in Catalonia*

Catalonia has gone through several phases of adaptation of the higher education model that is currently being implemented. These changes have been determined by the regulations applied by the Government of the Generalitat de Catalunya, essentially in three stages:

1.  Hybrid model : from the beginning of the course 20–21 until 30 October. In this first phase, teaching was organised on the basis of hybrid model criteria such as 50% of students attending in person and the remaining 50% attending online synchronously via video-call systems (Zoom or others). In this way, once every two weeks, 100% of the students attend the centre to carry out a practical activity, consisting of presentations and exams. This model was also reintroduced after the reduction of SARS-CoV-2 cases in Catalonia as of 8 February.
2.  Online synchronous model: from 1 to 30 November, as a reaction to the increase in the number of SARS-CoV-2 infections, the Catalan government decided to place the entire population in municipal perimeter confinement throughout Catalonia. From this moment on, classes become 100% synchronous.
3.  Synchronous and blended model: from 7 December 2020 until the end of the first semester, the Generalitat allows students to attend the centre from time to time to carry out practical activities. In this case, the synchronous teaching model is maintained while, at the same time, the different groups attend the centre on a scheduled basis to carry out certain practical activities.

The three scenarios described above have meant that teachers have had to readapt, modify and create different materials adapted to each specific moment.

For a better understanding of the situation in which teachers have had to face, the following Figure 2 shows the evolution of the number of cases of coronavirus diagnosed in Catalonia.

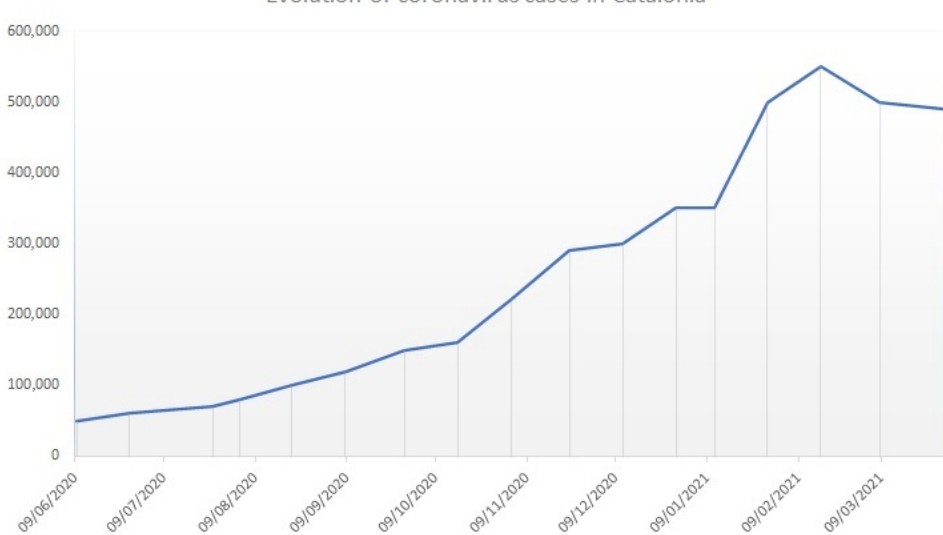

**Figure 2.** Evolution of coronavirus cases diagnosed in Catalonia (9 June 2020–21 March 2021). Source: EPDATA [33].

### 4.3. Transformative Learning by ESIC

Assuming the need for the competency-based model and putting the student at the centre of learning, ESIC Business and Marketing School developed a new teaching methodology that undertakes the change in teaching in two ways. On the one hand, as has been established, it is a model that combines the competency-based model and Student Center Learning, but at the same time accepts that today's society is digital. Transformative Learning by ESIC is inspired by the new ways of relating, living and working, breaking with traditional teaching models.

This new methodology is structured on a mixed and flexible training that combines remote attendance and physical attendance, turning the classroom and the campus into infinite spaces that combine the best of both worlds: the latest technologies and personal contact. All of this, guaranteeing the highest standards of teaching quality, as well as constant interaction with faculty, students and other areas of the school.

This model is based on the following axes:

1. Student Centered Learning.
2. Realistic and possible.
3. Technological.
4. Simple in its approach
5. Rapid in its implementation.
6. Innovative for the future.
7. Reliable: security and confidence with regards to families.
8. Teaching guarantee: for the teaching community.
9. Differentiating: for the market.

It is a model that reflects current needs and prepares students for the challenges they will face in an increasingly digital and competitive society.

As has been determined throughout the previous sections, one of the main issues to take into account when dealing with a situation such as the one experienced during the COVID-induced lockdown, is the response given by the institution.

In this sense, it is a matter of having a clear strategy that provides an appropriate working space for the teacher, but security for the student. It is clear that remote training is not the ideal situation as it limits/conditions the practice of activities, socialisation and teacher/student interaction, but it presents opportunities and also allows teaching to continue in situations where it is impossible to travel.

In this sense, ESIC established a series of protocols, as well as lines of work that allowed us to quickly and efficiently deal with the situation we were going through.

The central services worked to provide the necessary technology for the teaching staff. This allows us to distinguish two stages:

1. In the first stage, remote classes (given the confinement) were taught from the teachers' private homes. Therefore, in this first stage, software such as Zoom was provided for remote classes. At the same time, the use of Teams was extended and standardised at all levels.
2. In the second stage, work was done to incorporate hardware (touch screens, microphones, monitors) in the classrooms for training in a hybrid format, once the confinement was over.

From the outset, the institution assumed that the situation would require an alternative plan to traditional teaching, and instead of postponing the decision making process, work began from the outset on the foundations of the hybrid model.

This rapid response on the part of the school resulted in the following data:

1. The level of student attendance remained above 80%.
2. The level of student satisfaction as reflected in tutorials and delegate input was noticeably high and there was recognition of the effort and rapid adaptation of the university to remote teaching.
3. The programmes for all subjects were completed in accordance with the Teaching Guides and the objectives set were achieved.

However, as has been made clear previously, one of the keys to ESIC's transformation has not only been to focus on responding to the change in teaching during confinement, but also to assume the change of model by proposing a new way of understanding teaching. This resulting model has been called Transformative Learning at ESIC, based on the following specific aspects:

1. Cyclical model: On the one hand, the group is divided into two halves, two groups per class and on the other hand, there are 3 weeks per cycle. In each week something different happens for the pupil. Finally, the classroom is taken as the synchronous physical place of reference where "everything happens" and Canvas as the asynchronous digital place where preparations, information, knowledge encounter and learning management take place.
2. Compulsory attendance: It is obligatory, as in the current model. Attendance register is taken as normal. To those in class, in class, and to those at home, at home. Camera ON compulsory.
3. Dynamic programming: The beginning of the classes in this model is the beginning of the timer for learning and it is our obligation to lead the times and reaffirm this programming with the students. The scheduling of activities is the basis for guiding the learner through their transformation. It is also the way to hold the learner accountable. Student responsibility for their schedule of activities is essential to their academic experience. Our RIGOUR in the completion of scheduled activities is also the way in which the learner verifies his or her level of achievement or failure. A student who understands his/her errors is a student who has learned.
4. Classroom everywhere: To the extent that we assume that there are more spaces than the classroom for teaching, and we will be taking advantage of the model more. We leave the comfort zone of the classroom and understand that learning also happens, or can happen, in Canvas.
5. Self-consumption of knowledge: The teaching resources are at the student's disposal and the quantity or quality of their use will depend on the student. Readings, links, videos, tasks, mini-cases, collaborative content generation. All the tasks, activities and materials we provide must be consumed by the student for their learning process.

6. Assessment and feedback: The model requires a significant role for assessment and feedback on learning. The weight of individual work must increase, but participation and group work must, in turn, continue to be the basis of the student's transformation.

Teaching in this new ecosystem is marked by these three pillars:

1. Traditional teaching: The model is based on respect for the figure of the teacher.
2. The tech equipment of the classroom. Hardware and software. As has been pointed out in the previous points, technology is a fundamental part of today's teaching.
3. Innovative teaching resources.

### 4.4. Teaching Project Proposal

From what has been described above, it is easy to deduce where the main problem lies when it comes to teaching planning: uncertainty. The 2019–2020 academic year began as a face-to-face course and since 14 March it has been entirely online. In turn, during the 2020–2021 academic year in Catalonia, it began with hybrid planning and became synchronous online due to the advance of contagions; it then moved to a synchronous online model in which face-to-face practice was allowed and, finally, the hybrid model returns in the first years of each university degree course, a methodology that will be extended to the rest of the academic years in relation to the evolution of the contagions. In other words, four completely different methodologies in six months of the course.

This has led us to propose the following characteristics regarding teaching planning that we must take into consideration:

Educational planning/Teaching project.

- The PD is the result of a process of reflection on teaching activity.
- The PD plans the teaching activity and cannot have a closed or definitive character; it must be reviewed and updated with a view to optimising its future application.
- The PD should contain a contextualisation of the work scenario (institution, student body, resources, teaching culture of the institution, levels of transversal coordination, etc.).
- The PD should contain a reasoned statement of the author's teaching concept (university philosophy, educational values, attitudes, self-evaluation, teaching innovation).
- The PD must be useful to its author, to his/her teaching work1.

Along with this series of characteristics, the teacher has to assume that teaching has changed and that the face-to-face classroom model, where lectures prevail, cannot be transferred directly to the virtual classroom. Each space is different and, therefore, entails different ways of teaching. - Motivating active methodologies; online curricular practices; gamification and online games; asynchronous mechanisms to consolidate knowledge.

- Predictive systems for final assessment;
- Increasing digital competences.
- Improving educational innovation practice.

Among the most popular active learning strategies for classrooms are the flipped classroom and collaborative and cooperative learning.

The flipped classroom is a methodology that consists of "turning the traditional classroom upside down" and is increasingly used in the university context. Its use favours adaptation to a changing society and a new way of relating teachers and students [34]. New technologies favour the use of this methodology, with the student initiating the study of the content prior to attending the classroom and taking advantage of the subsequent meeting with the teacher to delve deeper into more complex issues. However, it also requires more commitment from the student [35,36].

On the other hand, collaborative and cooperative learning activities which, although they are often used as if they were the same, refer to different activities and each of them refers to different applications [37]. However, they have in common that, unlike individual learning, in these practices two or more people try to learn together in an interpersonal relationship that is conducive to the learning of the participants. They also

share the conception that knowledge can be created through the different interactions that are established between the different participants and the idea that learners and teachers are involved in a common task in which each depends on and is responsible for the other. With this type of learning, finally, there are interactions that can take place face-to-face, in an immediate face-to-face environment, or through computers in a remote environment using the tools of online forums, chat rooms, instant messaging, etc. [38].

## 5. Discussion

Area, San Nicolás and Sanabria, "the virtual classrooms of face-to-face teaching function more as an appendix or *ad hoc* support to the traditional teaching model, than as a revulsive or catalyst for pedagogical innovation in university teaching." [39].

The inclusion of ICT in education could enhance students' motivation, as well as their technological competences and skills [40].

In Spain, art. 81 of Organic Law 3/2018, of 5 December, on Personal Data Protection and guarantee of digital rights recognises the right to universal access to the internet, although it is not granted the status of a constitutionally proclaimed fundamental right.

From a teaching perspective it can be assumed that there have been changes and adaptations to the circumstances, such as the case study of ESIC Business and Marketing School proposed in this paper. We should reflect on the question posted by Archer: [22] "Were universities prepared to move from face-to-face to virtual teaching? Yes and no".

Although the change has indeed been abrupt and the leap from the face-to-face model to the hybrid model has taken place in a couple of weeks, based on the findings of our study, it could be argued that this is not a problem of the universities themselves but of the regulations imposed. As we analysed, educational institutions have done their best, also teachers [23], but different regulatory changes have been the real problem.

In an environment of uncertainty and change, a commitment to learning tools and strategies that facilitate rapid adaptation to change can allow teaching to continue regardless of whether the majority teaching system is face-to-face or online. In the teaching experiences derived from the health pandemic situation caused by COVID-19, Number of students enrolled in the activity it has been possible to show how the flipped classroom and collaborative and cooperative learning strategies have enabled teaching and learning to adapt to the demands imposed by the regulations of the different governments, while at the same time favouring student learning.

## 6. Conclusions

The methodological adaptation experienced in the 20–21 academic year due to the situation arising from the COVID-19 crisis has led university teachers to adapt in order to be able to continue teaching their subjects in ICT-mediated spaces.

In this scenario, it is important to open spaces for reflection and collaboration, and to give visibility to the best practices developed to further enhance creativity in didactic design and educational innovation.

In this respect, there is no doubt that the current trend is for lectures and face-to-face classes supported by slides projected in the classroom to give way to other methods in which student involvement and participation is greater. By way of example, it is sufficient to highlight the growing prominence of the Moodle or Canvas platforms. However, it is a fact that the exceptional situation caused by the declaration of the State of Alarm due to the COVID-19 pandemic has accelerated (or perhaps imposed) this phenomenon.

In this sense, during the pandemic, the need to digitalise the university and, more specifically, the teaching methods has become evident. Nowadays, all classrooms have an online replica, where students can consult all the information related to the subject.

In other words, the student has at all times a digital consultation space where he/she can see the main dates for deliveries and evaluations, the regulations relating to the work and, on many occasions, full explanations of what the exam will be like in order to facilitate its performance and avoid technical problems on the day it takes place. These

spaces have also relieved some of the traditional workload of attending classes and have allowed class dynamics to be transformed. In lecture classes, students used to spend a large part of their time taking notes. Now, given that many problems can arise from technology, such as problems in understanding what the teacher says (in some cases because sometimes the equipment fails or because the internet connection is not always the desired one), classes have become more dynamic. That is, students now have a large amount of material provided by the teacher prior to the lecture which allows them to follow the lecture with little or no note-taking. This enables them to be more attentive in class and more participative.

On the other hand, and this is something that provides added value, the classes in some centres, such as the case study presented here, are recorded. The student who attends the class can focus on carrying out the activity proposed by the teacher or participating in the debates and exercises proposed, as he/she has the peace of mind that, afterwards, he/she will be able to consult any specific doubts that may arise.

We expect the impact on the learning process to be very positive. According to our previous experience, the flipped classroom will increase students' participation in classroom sessions and their motivation for the subject.

Finally, and this is interesting, the Transformative Learning by ESIC model is not envisaged or designed as a one-off response to a specific problem, but has been implemented as a way of working to last over time. It is clear that during the period of confinement the break was abrupt. However, taking on teaching as a hybrid element between online and face-to-face allows students to develop a whole series of competences that will later serve them in their day-to-day work.

It provides the centre with a clear structure and idea of action, which allows it to organise spaces and resources in an efficient and stable way. Similarly, maintaining this kind of active teaching, where the teacher is obliged to keep the virtual classroom updated at all times, means that the leap to a 100% model for reasons of health or other emergencies is not traumatic.

Certainly, in other institutions with other social, personal and technological conditions, the responses to the legislative changes in higher education in Spain and Catalonia analysed would have been very different. Here we only analyse the case of ESIC, which makes it extremely difficult to extend these same conclusions to other contexts and circumstances. Future lines of research would probably have to analyse more universal and generalist models.

In short, teaching planning from now on must be digital, adaptable and permeable to the different scenarios that may occur, as has been shown in this study, and must contain sufficient resources to adapt to any of the possible scenarios that may arise. If our environment is dynamic, our teaching programmes, consequently, must also be dynamic.

**Author Contributions:** The article consists of five six sections: introduction, theoretical framework, methodology, results, discussion and conclusions. J.B.D. and T.M.V. have worked together in the elaboration of the theoretical framework and in the bibliographic search and compilation. R.N.-S. wrote the introduction and was the architect of the methodology. On the theoretical framework, J.B.D., T.M.V. and R.N.-S. worked together throughout the study. Regarding the results, the authors have worked collaboratively: J.B.D. has elaborated an initial draft on which T.M.V. and R.N.-S. have worked, contributing and making valuable contributions. Finally, both the discussions and the results have been part of the teamwork of the authors, J.B.D., T.M.V. and R.N.-S. All authors have read and agreed to the published version of the manuscript.

**Funding:** This research received no external funding.

**Institutional Review Board Statement:** Not applicable.

**Informed Consent Statement:** Not applicable.

**Data Availability Statement:** Not applicable.

**Acknowledgments:** This article has been developed within the Research Group on New Teaching Technologies of ESIC Business and Marketing School.

**Conflicts of Interest:** The authors declare no conflict of interest.

**Abbreviations**

The following abbreviations are used in this manuscript:

CRUE     Conference of Rectors of Spanish Universities
RD        Royal Decree-Law

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
