# Peer review of "University Teaching Planning in Times of COVID-19: Analysis of the Catalan Context and Proposal for a Future Model from ESIC Business and Marketing School Experience"

_sustainability, doi:10.3390/su13115936_

Round 1

Reviewer 1 Report

The study addresses important issues triggered by the COVID-19 pandemic to universities.  

My comments:
(1) research needs a problem, could you please state a problem that is solved by the paper
(2) Hypothesis needs a theoretical backbone. Based on what hypothesis was formulated (see page 3)?
(3) What about the validity and reliability of conducted qualitative content analysis?
(4) On page 3 you mention ".. effect on the university". How you measured it? What the "effect" means?
(5) Discussion is rather short. I would expect a more in-depth analysis and discussions support with theory.

Author Response

Dear reviewer,

Firstly, we would thank you very much for your comments and feedback: with any doubt they contribute to improve our paper.

We are very glad to answer to your kindly comments below them:

(1) research needs a problem, could you please state a problem that is solved by the paper

Regarding the research problem we propose how covid 19 has transformed the way of teaching. So, the study not only preambles the legal regulations that have been developed during the period, but also proposes the ESIC model (as a case study) as a long-term alternative solution.

(2) Hypothesis needs a theoretical backbone. Based on what hypothesis was formulated (see page 3)?

In addition to the previous answer, the hypothesis responds to two issues. Firstly, we are not only talking about a hypothesis based on theoretical knowledge, but one that arises from a global situation such as that produced by covid 19. On this basis, the hypothesis takes up, or aims to take up, the sum of all the issues raised by recent literature on education where the face-to-face model is questioned, and the shortcomings derived from the hybrid model are analysed. This is a fact that we have tried to include as comprehensively as possible in the theoretical framework. At any way and given the magnitude of the phenomenon of this pandemic, the number of publications on this subject has increased substantially, so that the theoretical elaboration has been limited to a specific framework.

(3) What about the validity and reliability of conducted qualitative content analysis?

Regarding the sample, as indicated in the study phases, a compilation of all the regulations published in the territories has been made analysed. We are aware that each of Spain's autonomous communities has initiated its own procedures, protocols and regulations. For this reason, our study was restricted to Catalonia only. On this basis, this study has explored all the regulations on the issue since the proclamation of the state of alarm.

(4) On page 3 you mention ".. effect on the university". How you measured it? What the "effect" means?

Maybe the correct term has not been applied here. It would be more appropriate, as the reviewer has point out, to use the word "application" to avoid misunderstandings as the reviewer considered. We consequently changed this point.

(5) Discussion is rather short. I would expect a more in-depth analysis and discussions support with theory.

Comment considered. A more in-depth analysis was introduced in the paper as reviewer suggested.

Thanks again for your comments and feedback.

We remain at your disposal if required.

Best wishes,

The authors.

Reviewer 2 Report

Regarding the content, I do not have any changes to recommend, it makes a good literary review to support the relevance of the problem to be studied and a good structuring of the content, it uses the correct methodology for this type of study and it is a consistent and well-detailed methodology to give significance to the results they show, makes a good discussion of the results with respect to the studies carried out previously, and marks the conclusion obtained well.

Once the authors have made these changes, I think the article is publishable.

The title should refer to ESIC, since part of the analysis is based on that university.

There can be no citations in the abstract (lines 2 and 5).

A table or figure should never appear without reference in the text prior to it (you have table 1 and figure 2 without referencing).

The sources of figures 1 and 2 must be accompanied by the numerical citation and have their reference in the section ‘References’.

And in the sixth section, you should add a paragraph at the end that talks about the limitations and includes deductions for future research.

Author Response

Dear reviewer,

Firstly, we would thank you very much for your comments and feedback: with any doubt they contribute to improve our paper.

We are very glad to answer to your kindly comments below them:

  1. The title should refer to ESIC, since part of the analysis is based on that university.

Title modified as reviewer suggested.

  1. There can be no citations in the abstract (lines 2 and 5).

Citations removed as reviewer suggested.

  1. A table or figure should never appear without reference in the text prior to it (you have table 1 and figure 2 without referencing).

Table 1 is referred at line 174, in the following text:

“According to data provided by the Ministry of Health, the last day on which Spain had 0 SARS-CoV-2 infections was 24 February 2020. Eighteen days later, the number of daily cases diagnosed in Spain (Figure 1) was 1,708 persons.”

The reference for the Table 2 is added as reviewer suggested.

  1. The sources of figures 1 and 2 must be accompanied by the numerical citation and have their reference in the section ‘References’.

References were added in the section ‘References’ as reviewer suggested.

  1. And in the sixth section, you should add a paragraph at the end that talks about the limitations and includes deductions for future research.

Paragraph with the limitations and further research added at sixth section, as reviewer suggested.

  1. A table or figure should never appear without reference in the text prior to it (you have table 1 and figure 2 without referencing).

Same answer at question 3.

Thanks again for your comments and feedback.

We remain at your disposal if required.

Best wishes,

The authors.

Reviewer 3 Report

I am happy with this version of the paper now. The English is much better. The study is clear and makes a good contribution to how the COVI 19 pandemic has provided tertiary education in Catalonia with new opportunities. Online education is becoming very much mainstream in Universities now.

More detailed comments:

The qualitative method is explained well.

This is a good case study of a response to COVID in university education in a region that had not traditionally engaged more fully with online education.

Line 79: the word had is spelt as h ad

Line 200: “As a complement to this information…..” – the meaning of the word “complement” is unclear. Can the authors used a clearer word?

Line 293: “confinement resulting from the state of alarm” sounds clumsy. It is better to write “COVID-induced lockdown”.

In the Conclusion, I notice the phrase “On the other Hand” is used twice. It would be better for readability if this phrase was used just once and a different phrased used instead. For example, However.

Author Response

Dear reviewer,

Firstly, we would thank you very much for your comments and feedback: with any doubt they contribute to improve our paper.

We are very glad to answer to your kindly comments below them:

  • Line 79: the word had is spelt as h ad

Spelt correct as reviewer suggested.

  • Line 200: “As a complement to this information…..” – the meaning of the word “complement” is unclear. Can the authors used a clearer word?

A new clearer expression is used in the text as reviewer suggested.

  • Line 293: “confinement resulting from the state of alarm” sounds clumsy. It is better to write “COVID-induced lockdown”.

Change done as reviewer suggested.

  • In the Conclusion, I notice the phrase “On the other Hand” is used twice. It would be better for readability if this phrase was used just once and a different phrased used instead. For example, However.

 A new expression is used in the text as reviewer suggested.

Thanks again for your comments and feedback.

We remain at your disposal if required.

Best wishes,

The authors.

Round 2

Reviewer 1 Report

Thank you for the updates and improvements the authors have made. I appreciate what has been done.

However, I have an additional comment. I still do not understand why you use "hypothesis" (line 108). I should say it is now more like a problem claim. The hypothesis needs theoretical justification.

Author Response

Dear reviewer,

So many thanks for your comments and recognition.

We  had considered the question you pointed up and, certainly, for avoiding any misunderstanding about this point, we followed your interesting suggestion. That is why we removed the word "hypothesis" from the text and we defined it as problem claim.

Thanks again for your comments they help us to improve the paper.

Kind regards,

Teodor Mellen